# BIAS AS A VIRTUE: RETHINKING GENERALIZATION UNDER DISTRIBUTION SHIFTS

## ABSTRACT

Machine learning models often degrade when deployed on data distributions different from their training data. Challenging conventional validation paradigms, we demonstrate that higher in-distribution (ID) bias can lead to better out-of-distribution (OOD) generalization. Our Adaptive Distribution Bridge (ADB) framework implements this insight by introducing controlled statistical diversity during training, enabling models to develop bias profiles that effectively generalize across distributions. Empirically, we observe a robust negative correlation where higher ID bias corresponds to lower OOD error—a finding that contradicts standard practices focused on minimizing validation error. Evaluation on multiple datasets shows our approach significantly improves OOD generalization. ADB achieves robust mean error reductions of up to 26.8% compared to traditional cross-validation, and consistently identifies high-performing training strategies, evidenced by percentile ranks often exceeding 83.4%. Our work provides both a practical method for improving generalization and a theoretical framework for reconsidering the role of bias in robust machine learning.

## 1 INTRODUCTION

Machine learning models frequently suffer performance degradation when deployed in environments with distributions different from their training data (Moreno-Torres et al., 2012; Taori et al., 2020). This distribution shift problem is particularly acute in scientific discovery domains where training data encompasses only limited experimental spaces, or real-world applications like finance where historical data often differs systematically from future market conditions (Koh et al., 2021).

Traditional approaches to model selection rely on cross-validation, which implicitly assumes that models with lower in-distribution (ID) error will generalize better to out-of-distribution (OOD) settings. However, mounting evidence suggests this assumption fails under significant distribution shifts (Sagawa et al., 2020; Gulrajani & Lopez-Paz, 2021), creating a critical gap between theoretical expectations and practical performance.

We challenge this foundational practice by demonstrating that models with strategically increased ID bias can achieve significantly better OOD generalization. Our theoretical analysis reveals that when the bias diversity is appropriately calibrated relative to the distribution shift magnitude, the correlation between ID and OOD errors becomes negative. In this counterintuitive regime, conventional model selection criteria that minimize ID error paradoxically select models with worse OOD performance.

Building on this insight, we introduce the Adaptive Distribution Bridge (ADB) framework, which leverages controlled statistical diversity during training to enhance OOD generalization. ADB quantifies and selects training sequences using debiased optimal-transport distances, inducing appropriate bias variation without requiring explicit knowledge of target distributions. Our approach creates models whose inherent bias profiles effectively navigate distribution shifts while maintaining focus on invariant patterns.

Experiments across multiple domains show ADB achieves substantial mean OOD error reductions, for instance, by up to 26.8% in MAE on the T1S1 dataset using the Cumulative method (Table 1), when compared to traditional cross-validation. Our approach consistently identifies high-performing

training strategies (with percentile ranks often above 83.4%). These findings validate our theoretical framework and offer a practical alternative to established validation paradigms.

**Contributions:** (1) We establish theoretical conditions under which higher ID bias leads to reduced OOD error; (2) We introduce the ADB framework implementing controlled statistical diversity during training; (3) We demonstrate empirically that our approach significantly outperforms standard cross-validation on multiple prediction tasks; (4) We establish a paradigm that reconceptualizes bias as a potential virtue for addressing distribution shifts.

## 2 RELATED WORK

### 2.1 BIAS-VARIANCE TRADEOFF AND MODEL SELECTION

The bias-variance tradeoff represents a fundamental concept in statistical learning theory (Domingos, 2000; James, 2003). Conventional model selection minimizes validation error as a proxy for generalization performance, underpinning cross-validation (Arlot & Celisse, 2010) across machine learning applications.

Recent work questions this orthodoxy in specific contexts. D'Amour et al. (2020) highlighted how models with equivalent validation performance can behave drastically differently under distribution shifts. Kumar et al. (2022) demonstrate that minimizing in-distribution error during fine-tuning can actively harm OOD performance, providing direct evidence for our thesis. Efforts to improve robustness include seeking invariant features across environments (Arjovsky et al., 2019), employing ensemble methods for bias diversity (Teney et al., 2022), and leveraging feature-level diversity (Izmailov et al., 2022), all of which contribute to reconsidering bias as a potential virtue rather than a pure detriment.

Our work builds upon these insights by proposing a novel framework that, instead of solely focusing on bias minimization or feature invariance, strategically manages and leverages controlled bias variation to enhance generalization under distribution shifts.

### 2.2 ADDRESSING DISTRIBUTION SHIFTS

Existing approaches to distribution shifts include: domain adaptation methods (Ganin et al., 2016; Tzeng et al., 2017) that align feature distributions between source and target domains; robust optimization approaches (Sagawa et al., 2019; Duchi & Namkoong, 2018) that optimize for worst-case performance; and causality-based techniques (Peters et al., 2016; Schölkopf et al., 2021) that identify invariant causal mechanisms.

A key limitation across these approaches is their dependence on specific knowledge about target distributions or precise characterization of shift types. Recent benchmarks and evaluations (Wiles et al., 2022; Santurkar et al., 2021; Shen et al., 2021) demonstrate that many existing methods fail to generalize reliably across diverse real-world distribution shifts. ADB addresses these limitations by introducing controlled diversity during training without requiring explicit knowledge of target distributions.

### 2.3 TRAINING DYNAMICS AND SEQUENCE MANIPULATION

How models interact with training data—particularly the order and composition of training batches—significantly impacts learning outcomes. Curriculum learning (Bengio et al., 2009; Graves et al., 2017) arranges examples by difficulty, while active learning (Settles, 2009; Sener & Savarese, 2018) iteratively selects informative examples. Recent work by Mindermann et al. (2022) demonstrates the benefits of prioritizing training on points that are learnable, worth learning, and not yet learned, showing how strategic data selection affects generalization. Killamsetty et al. (2021) propose gradient-matching approaches for efficient training data subset selection, further establishing the importance of data presentation strategies.

ADB differs from these approaches by focusing on distributional characteristics of entire training sequences rather than example-level properties. Unlike methods that typically assess individual examples or local batch compositions, ADB uniquely quantifies the evolving distributional character-

istics of training subsequences derived from permutations in relation to the overall distribution of the training dataset. While previous work has observed how training order affects generalization (Hacohen & Weinshall, 2019), ADB provides a theoretical foundation linking sequence diversity to bias profiles and explicitly quantifies these relationships using optimal transport distances. This enables systematic induction of controlled bias variation beneficial for OOD generalization, aligning with recent data-centric AI perspectives (Zha et al., 2023) that emphasize the central role of data management in model performance.

# 3 THEORETICAL FOUNDATION: BIAS UNDER DISTRIBUTION SHIFTS

## 3.1 MEAN SHIFT MODEL

We analyze a simplified model where error metrics in training and test distributions are generated as:

$$\mathcal{E}_{\text{train}} = \mu_{\text{train}} + \epsilon, \quad \epsilon \sim \mathcal{N}(0, \sigma_{\text{train}}^2), \tag{1}$$

$$\mathcal{E}_{\text{test}} = \mu_{\text{test}} + \epsilon', \quad \epsilon' \sim \mathcal{N}(0, \sigma_{\text{test}}^2), \tag{2}$$

with a positive mean shift $\Delta = \mu_{\text{test}} - \mu_{\text{train}} > 0$, representing the common scenario where models perform worse on test data than training data due to distribution shifts.

We consider candidate models whose expected error deviates from the training mean $\mu_{\text{train}}$ by a bias parameter $b$. Traditional cross-validation prioritizes models with $b$ near zero to minimize ID error. However, when $b = 0$, with mean square error metric, the OOD error becomes $\Delta^2$, which can be substantial when the shift $\Delta$ is significant.

## 3.2 CORRELATION BETWEEN ID AND OOD ERRORS

For a model with bias parameter $b$, we define $T(b) = b^2$ (ID error component) and $U(b) = (b - \Delta)^2$ (OOD error component).

We model $b$ as following a half-normal distribution with scale parameter $k$. Let $\alpha = \sqrt{\frac{2}{\pi}}$. The half-normal probability density function is:

$$f(b; k) = \begin{cases} \frac{\alpha}{k} \exp\left(-\frac{b^2}{2k^2}\right), & \text{if } b \geq 0 \\ 0, & \text{if } b < 0 \end{cases} \tag{3}$$

where $k$ represents bias variation across model populations.

We select the half-normal distribution for three key reasons: (1) it restricts the bias parameter to non-negative values, reflecting the practical scenario where we eliminate underfitting models during training; (2) it provides a single parameter $k$ that directly controls the spread of bias variation, enabling clear analysis of the relationship between bias diversity and OOD performance; and (3) it has well-understood statistical properties that allow for closed-form derivation of the correlation between ID and OOD errors.

Through mathematical derivation (Appendix A), we prove that the Pearson correlation coefficient $\rho_{T,U}$ between $T(b)$ and $U(b)$ becomes negative when $k < \alpha\Delta$.

This condition explains both standard (non-shifting) and distribution shift scenarios. In conventional i.i.d. settings ($\Delta \approx 0$), the inequality $k < \alpha\Delta$ is virtually impossible to satisfy since $k > 0$ by definition, explaining the typical positive correlation between ID and OOD errors. However, under significant distribution shifts where $\Delta$ is large, the condition $k < \alpha\Delta$ becomes feasible to satisfy, enabling negative correlation.

## 3.3 OPTIMIZING THE BIAS-DIVERSITY TRADEOFF

When $k < \alpha\Delta$, the correlation $\rho_{T,U}$ is negative. As $k$ increases within this range, $\rho_{T,U}$ increases toward zero, meaning the negative correlation weakens. Therefore, the most negative correlation occurs at the smallest feasible values of $k$.

This creates a practical tradeoff: we need $k$ large enough to provide sufficient bias diversity for distribution shift compensation, yet small enough to maintain a strong negative correlation between ID and OOD errors.

The critical challenge is achieving this appropriate level of bias variation $k$ in practice, which motivates our ADB framework.

# 4 ADAPTIVE DISTRIBUTION BRIDGE (ADB) FRAMEWORK

## 4.1 FRAMEWORK MOTIVATION AND OVERVIEW

Building on our theoretical finding that appropriate bias variation $k < \alpha\Delta$ improves OOD performance through negative ID-OOD error correlation, the ADB framework controls training sequences to indirectly influence $k$. We achieve this by quantifying the distributional characteristics of potential training sequences, categorizing these sequences based on their statistical diversity, and identifying sequences that induce appropriate bias variation for effective OOD generalization. The core intuition is that by exposing the model to training sequences with varying degrees of statistical divergence from the global data distribution, we can prevent premature convergence to a narrow set of features and thereby modulate the model's learned bias profile, akin to influencing the bias variation parameter $k$ from our theoretical model.

## 4.2 PROBLEM FORMULATION

Let $\mathcal{D} = \{(x_i, y_i)\}_{i=1}^{N}$ be a dataset of $N$ samples. We train a model $f_\theta : \mathcal{X} \to \mathcal{Y}$ using mini-batch stochastic gradient descent. A permutation $\pi$ defines a training sequence, with batches defined as:

$$B_t^\pi = \left\{ (x_{\pi((t-1)B+j)}, y_{\pi((t-1)B+j)}) : j = 1, 2, \ldots, \min(B, N - (t-1)B) \right\} \quad (4)$$

where $B$ is the batch size and $t$ indexes the training step.

Our objective is to identify permutations that introduce the appropriate level of statistical diversity to induce beneficial bias variation $k$ that enhances model robustness under distribution shifts.

## 4.3 DISTRIBUTIONAL DIVERSITY QUANTIFICATION

To quantify distributional diversity, we represent data samples in a space where distributional distances are meaningful, measure the deviation between training batches (or cumulative subsets) and the global training distribution, and categorize permutations based on their deviation patterns.

For high-dimensional problems, we employ a Variational Autoencoder (VAE) (Kingma & Welling, 2013) to map features to a lower-dimensional latent space for efficient distance computation.

We quantify statistical divergence using the Sinkhorn distance (Cuturi, 2013), which provides a computationally tractable approximation to the Wasserstein-1 distance ($W_1$):

$$W_\epsilon(\mu_S, \mu_G) = \langle \mathbf{C}, \mathbf{P}^* \rangle \quad (5)$$

where $\mathbf{C}$ is the cost matrix with elements $C_{ij} = \|z_i - z_j\|_1$ and the optimal transport plan $\mathbf{P}^*$ is:

$$\mathbf{P}^* = \arg \min_{\mathbf{P} \in \Pi(\mu_S, \mu_G)} \left\{ \langle \mathbf{C}, \mathbf{P} \rangle - \epsilon H(\mathbf{P}) \right\} \quad (6)$$

Here, $\mu_S$ and $\mu_G$ represent the empirical probability distributions of a sample subset and the global training set, respectively. $\Pi(\mu_S, \mu_G)$ is the set of all transport plans between these distributions, and $H(\mathbf{P}) = -\sum_{i,j} P_{ij} \log P_{ij}$ is the entropy of the transport plan.

To mitigate entropic bias, we compute the debiased distance:

$$W_{\text{debiased}}(\mu_S, \mu_G) = 2W_\epsilon(\mu_S, \mu_G) - W_\epsilon(\mu_S, \mu_S) - W_\epsilon(\mu_G, \mu_G) \quad (7)$$

Algorithm 1 summarizes the resulting Adaptive Distribution Bridge workflow.

---

**Algorithm 1** ADB Framework

---

**Require:** Dataset latent representations $\{z_i\}_{i=1}^N$, batch size $B$, permutation count $M$, mode $\in$ {cumulative, batchwise}
**Ensure:** Permutation sets $\Pi_{\text{low}}, \Pi_{\text{med}}, \Pi_{\text{high}}$
 1: Generate $M$ random permutations $\{\pi_1, \pi_2, \ldots, \pi_M\}$
 2: $T \leftarrow \lceil N/B \rceil$
 3: **for** $m = 1$ to $M$ **do**
 4:    **for** $t = 1$ to $T$ **do**
 5:       **if** mode = cumulative **then**
 6:          $S_t^{\pi_m} \leftarrow \{z_{\pi_m(1)}, \ldots, z_{\pi_m(tB)}\}$
 7:          $D_t^{\pi_m} \leftarrow W_{\text{debiased}}(\mu_{S_t^{\pi_m}}, \mu_G)$
 8:       **else**
 9:          $B_t^{\pi_m} \leftarrow \{z_{\pi_m((t-1)B+1)}, \ldots, z_{\pi_m(\min(tB,N))}\}$
10:          $D_t^{\pi_m} \leftarrow W_{\text{debiased}}(\mu_{B_t^{\pi_m}}, \mu_G)$
11:       **end if**
12:    **end for**
13: **end for**
14: **for** $t = 1$ to $T$ **do**
15:    $\mu_{D,t} \leftarrow \frac{1}{M} \sum_{m=1}^M D_t^{\pi_m}$
16:    $\sigma_{D,t} \leftarrow \sqrt{\frac{1}{M} \sum_{m=1}^M (D_t^{\pi_m} - \mu_{D,t})^2}$
17: **end for**
18: **for** $m = 1$ to $M$ **do**
19:    $O^{\pi_m} \leftarrow \sum_{t=1}^T \mathbf{1}\big(D_t^{\pi_m} \notin [\mu_{D,t} - 2\sigma_{D,t}, \mu_{D,t} + 2\sigma_{D,t}]\big)$
20: **end for**
21: Determine thresholds $\tau_{\text{low}}$ and $\tau_{\text{high}}$ using distribution quantiles
22: $\Pi_{\text{low}} \leftarrow \{\pi_m : O^{\pi_m} \leq \tau_{\text{low}}\}$
23: $\Pi_{\text{med}} \leftarrow \{\pi_m : \tau_{\text{low}} < O^{\pi_m} \leq \tau_{\text{high}}\}$
24: $\Pi_{\text{high}} \leftarrow \{\pi_m : O^{\pi_m} > \tau_{\text{high}}\}$
25: **return** $\Pi_{\text{low}}, \Pi_{\text{med}}, \Pi_{\text{high}}$

---

### 4.4 COMPUTATIONAL APPROACHES

We introduce two approaches for computing distributional deviations:

**Cumulative Evaluation** This approach evaluates the distance between the cumulative subset of samples seen up to step $t$ and the global training distribution:

$$D_t^\pi = W_{\text{debiased}}(\mu_{S_t^\pi}, \mu_G), \text{ where } S_t^\pi = \{\pi(1), \pi(2), \ldots, \pi(tB)\} \tag{8}$$

**Batchwise Computation** This approach computes the deviation of individual batches from the global training distribution:

$$D_t^\pi = W_{\text{debiased}}(\mu_{B_t^\pi}, \mu_G) \tag{9}$$

### 4.5 PERMUTATION CLASSIFICATION

For each time step $t$, we compute the mean $\mu_{D,t}$ and standard deviation $\sigma_{D,t}$ of the distributional deviations across all permutations. We then count the number of times each permutation's deviation falls outside the $\pm 2\sigma$ bounds:

$$O^{\pi_m} = \sum_{t=1}^T \mathbf{1}\left(D_t^{\pi_m} \notin [\mu_{D,t} - 2\sigma_{D,t}, \mu_{D,t} + 2\sigma_{D,t}]\right) \tag{10}$$

Based on these counts, we classify permutations into three categories:

- Low deviation ($O^\pi \leq \tau_{\text{low}}$): Maintains tight alignment with the global training distribution
- Medium deviation ($\tau_{\text{low}} < O^\pi \leq \tau_{\text{high}}$): Provides moderate statistical diversity

- High deviation ($O^\pi > \tau_{\text{high}}$): Exhibits substantial distributional deviations

We determine thresholds $\tau_{\text{low}}$ and $\tau_{\text{high}}$ using distribution quantiles of the empirical distribution of outlier counts ($O^\pi$). We observed distinct patterns for the two computational approaches across datasets. For the Cumulative method, this empirical distribution consistently exhibits a pattern resembling exponential decay for low outlier counts, while the high end consists primarily of outlier values. For the Batchwise method, the empirical distribution approximates a normal distribution. The probability distribution across deviation groups and specific threshold values for both approaches are detailed in Appendix B, Table 3.

## 5 EXPERIMENTAL RESULTS

### 5.1 EXPERIMENTAL SETUP

We evaluate our approach on three datasets: T1S1 (molecular property prediction; currently a private dataset, planned for public release), QM9 (Ramakrishnan et al., 2014) (quantum chemistry with 134k molecules, using its zero-point vibrational energy (ZPVE) property for prediction), and Year Prediction MSD (Bertin-Mahieux et al., 2011) (audio features from 515k songs).

To evaluate generalization under distribution shifts, OOD test sets are created using stratified sampling. Training and test sets are normalized separately, resulting in $W_1$ distances in label space between OOD distributions ranging from 0.61 to 0.86, ensuring meaningful distribution shifts (detailed $W_1$ measurements in Appendix B). For instance, the T1S1 dataset (564,220 molecules) has its OOD test set (64,220 molecules) formed via this stratified sampling. The remaining 500,000 molecules are divided into a 490,000-sample training set and a 10,000-sample in-distribution (ID) validation set; the validation set is kept small to minimize disruption to training distribution properties. For both QM9 and MSD datasets, we use 90,000 samples for training, 10,000 samples for ID validation, and 10,000 samples for OOD testing, with all sets formed through the same stratified sampling approach.

Our ADB implementation processes the training samples as detailed in Algorithm 1 (with $M = 500$ permutations) to generate and classify these permutations into three deviation categories (Low, Medium, High) based on their statistical properties. Each category is evaluated by training 30 models per deviation group across each dataset. During training, each model uses 50 permutation sequences sampled from its assigned deviation category, with each sequence determining the training sample order for one epoch. This approach ensures consistent exposure to the designated level of statistical diversity throughout training.

The classification thresholds $\tau_{\text{low}}$ and $\tau_{\text{high}}$, and the probability distribution across deviation groups, are detailed in Appendix B, Table 3.

The experimental protocol is as follows:

- **Optimization**: Models are trained using Adam (Kingma & Ba, 2014) with learning rate 0.001 for 50 epochs.

- **Batch sizes**: We use batch size $B$ of 5000 for T1S1, 1000 for QM9, and 1000 for MSD (corresponding to the batch size parameter in ADB), each representing approximately 1% of their respective training set sizes.

- **Architecture**: For molecular datasets (T1S1 and QM9), we use message-passing neural networks following Gilmer et al. (2017), while MSD uses fully-connected networks.

- **Dimensionality reduction for ADB**: To facilitate distance computations in the ADB framework, Variational Autoencoders (VAEs) are employed. For molecular datasets (T1S1 and QM9), VAEs map 2098 input features (constructed from Morgan fingerprints (Morgan, 1965) and RDKit molecular properties (Landrum et al., 2013)) to a 32-dimensional latent space using a six-hidden-layer encoder architecture with power-of-2 dimension reduction. For the MSD dataset, VAEs map 90 input features to an 8-dimensional latent space via a three-hidden-layer encoder.

- **Optimal transport**: Sinkhorn algorithm with entropic regularization parameter $\epsilon = 0.05$.

Table 1: Performance evaluation of the ADB approach

| Method | Dataset | Metric | CV ↓ | ADB ↓ | I (%)↑ | PR (%)↑ |
|--------|---------|--------|------|-------|--------|---------|
| Batchwise | T1S1 | MAE | $0.478 \pm 0.018$ | $0.427 \pm 0.007$ | 10.7 | 83.4 |
| | | RMSE | $0.561 \pm 0.013$ | $0.521 \pm 0.007$ | 7.2 | 76.9 |
| | QM9 | MAE | $0.158 \pm 0.011$ | $0.136 \pm 0.001$ | 14.2 | 86.7 |
| | | RMSE | $0.217 \pm 0.031$ | $0.190 \pm 0.002$ | 12.7 | 67.8 |
| | MSD | MAE | $0.868 \pm 0.005$ | $0.854 \pm 0.001$ | 1.55 | 85.2 |
| | | RMSE | $1.163 \pm 0.007$ | $1.149 \pm 0.002$ | 1.22 | 70.5 |
| Cumulative | T1S1 | MAE | $0.514 \pm 0.020$ | $0.365 \pm 0.003$ | 26.8 | 91.2 |
| | | RMSE | $0.554 \pm 0.017$ | $0.459 \pm 0.003$ | 17.2 | 85.7 |
| | QM9 | MAE | $0.160 \pm 0.005$ | $0.129 \pm 0.001$ | 19.2 | 87.8 |
| | | RMSE | $0.202 \pm 0.011$ | $0.178 \pm 0.002$ | 11.9 | 75.6 |
| | MSD | MAE | $0.874 \pm 0.001$ | $0.850 \pm 0.001$ | 2.96 | 91.2 |
| | | RMSE | $1.159 \pm 0.005$ | $1.138 \pm 0.001$ | 1.81 | 91.1 |

Performance is assessed using Mean Absolute Error (MAE) and Root Mean Square Error (RMSE), with 10-fold cross-validation as the benchmark.

## 5.2 PERFORMANCE COMPARISON

Table 1 presents the effectiveness of our ADB framework compared to standard 10-fold cross-validation (CV). The percentage improvement (I) quantifies ADB's relative advantage over CV, while the percentile rank (PR) positions the ADB model within the distribution of all possible training permutations, with higher percentages indicating better relative performance. All performance improvements are statistically significant ($p < 0.05$) as demonstrated through paired t-tests (see Appendix B.4).

The results demonstrate a clear advantage for ADB over traditional CV. The Batchwise approach shows significant improvements in both MAE and RMSE, while the Cumulative approach delivers even stronger improvements with metrics ranking above the 75th percentile. The Cumulative approach consistently outperforms Batchwise because it accounts for the full historical sequence of training samples rather than isolated batches.

Notably, MAE improvements consistently exceed RMSE gains across both methods, demonstrating MAE's superior statistical robustness under distribution shifts due to its reduced sensitivity to outliers.

It is also worth noting that while the absolute percentage improvement (I) varies across datasets due to their distinct characteristics, the percentile rank (PR) remains consistently high, particularly for the Cumulative approach. This suggests that ADB reliably identifies high-performing permutation strategies relative to the entire space of possibilities, regardless of the specific dataset.

Our ADB framework favors Medium and High deviation groups, as these typically provide greater statistical diversity beneficial for robust OOD generalization. For Table 1, we sample 10 models proportionally from the Medium and High groups, select the one with the highest ID error, and report its OOD performance. Performance data for all deviation groups is available in Appendix B.3.

## 5.3 DISTRIBUTION-ERROR RELATIONSHIP ANALYSIS

Our analysis demonstrates that under negative correlation between ID and OOD errors (when $k < \alpha\Delta$), standard cross-validation that minimizes ID error leads to suboptimal OOD performance. In such cases, models with higher ID error can achieve lower OOD error.

Based on the negative ID-OOD correlation patterns observed in Figure 1, we focus on Medium and High deviation groups for model selection. By selecting the highest ID error model from these groups, ADB leverages this correlation pattern to improve OOD performance.

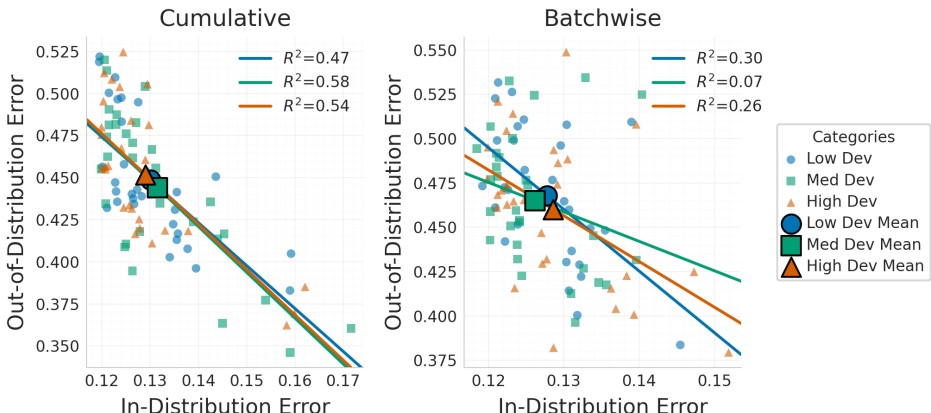

Figure 1: Relationship between in-distribution and out-of-distribution MAE across Low, Medium, and High deviation groups for T1S1 with both computational approaches. The negative correlation is evident, with distinct patterns for each group that reveal the underlying bias-diversity tradeoff.

This empirical finding aligns with our theoretical model (Section 3), which predicts that when $k < \alpha\Delta$, increased bias diversity can lead to improved OOD outcomes through negative ID-OOD correlation. The High deviation group exhibits larger $k$ (more bias diversity) but potentially weaker correlation strength, while the Medium group provides moderate $k$ with potentially stronger negative correlation. ADB's selection of the maximum ID error model effectively navigates this tradeoff without requiring precise knowledge of the optimal bias diversity level.

### 5.4 COMPUTATIONAL EFFICIENCY

The batchwise approach achieves substantial computational speedup compared to the cumulative approach, but at a significant performance cost as evident in our experimental results.

For computational complexity, let $N$ be sample count and $B$ be batch size. With the Sinkhorn algorithm for optimal transport having complexity $O(nm \log(nm)/\epsilon^2)$ for distributions with $n$ and $m$ points (Luo et al., 2023), where $\epsilon$ controls the entropic regularization:

$$C_{\text{cumulative}} = \sum_{t=1}^{N/B} O\left((tB)N \log(tBN)/\epsilon^2\right) = O\left(\frac{N^3 \log N}{B\epsilon^2}\right) \quad (11)$$

$$C_{\text{batchwise}} = \sum_{t=1}^{N/B} O\left(BN \log(BN)/\epsilon^2\right) = O\left(\frac{N^2 \log N}{\epsilon^2}\right) \quad (12)$$

With $B = 0.01N$, this yields a theoretical upper bound on speedup of approximately $100\times$. In practice, our empirical observations show speedups between $1.9\times$ and $4.1\times$ due to implementation considerations. For computational efficiency, our permutation analysis leverages parallel computation, with Monte Carlo simulations achieving linear speedup across multiple GPU resources.

For our largest dataset (500,000 samples), processing all 500 permutation paths required 266.5 total GPU hours with the batchwise approach versus 740 hours with the cumulative approach across 5 H800 GPUs.

## 6 DISCUSSION AND CONCLUSION

Our work demonstrates that higher ID bias can improve OOD generalization, challenging conventional validation paradigms. The ADB framework provides a practical implementation of this principle through controlled statistical diversity during training.

Key implications include:

Table 2: ADB computational efficiency comparison on H800 GPUs (GPU hours per permutation)

| Dataset Size | Method | Computation Time | Relative Speedup |
|---|---|---|---|
| $100,000 \times 8$ | Cumulative | 0.131 | 1.0× |
| | Batchwise | 0.070 | 1.9× |
| $100,000 \times 32$ | Cumulative | 0.151 | 1.0× |
| | Batchwise | 0.037 | 4.1× |
| $500,000 \times 32$ | Cumulative | 1.480 | 1.0× |
| | Batchwise | 0.533 | 2.8× |

- **Validation paradigm shift:** Traditional approaches that minimize ID error may harm generalization under distribution shifts. Our results demonstrate that cross-validation can lead to substantially higher OOD error, while our ADB framework yields significant mean improvements and consistently identifies high-performing strategies.

- **Practical framework:** ADB enhances model robustness while requiring only an awareness of distribution shifts within a reasonable range, without requiring detailed knowledge of their specific characteristics.

- **Theoretical foundation:** Our analysis linking ID bias and OOD error provides a basis for rethinking bias in machine learning.

Limitations and future work include further theoretical development, investigating how dataset characteristics and computational approaches influence the quantitative relationship between induced distributional deviation levels and the resulting bias-diversity tradeoff. The distributional diversity principles introduced in this work could inform active learning pipelines for AI-driven scientific discovery, where experimental costs constrain labeling budgets and query sequences must balance exploration of uncertain regimes with exploitation of known invariants. Integrating ADB-style permutation scouting with acquisition functions could help molecular design, materials discovery, or climate modeling systems prioritize batches that expose models to scientifically meaningful distribution shifts while respecting laboratory cadence. Code for the ADB framework and experiments will be made publicly available upon publication.

## REPRODUCIBILITY STATEMENT

Section 5 details the datasets, model architectures, optimization hyperparameters, and permutation sampling procedures required to reproduce our experiments. Public datasets (QM9 and MSD) are cited with acquisition details, and the private T1S1 dataset will be released with the camera-ready submission. We will publish the full ADB implementation, permutation generation utilities, and scripts for recreating all experiments, figures, and tables upon acceptance. Evaluation protocols are fixed and enumerated in Appendix B to facilitate exact replication.

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

## A   MATHEMATICAL ANALYSIS OF NEGATIVE CORRELATION

This appendix provides the mathematical derivation of the negative correlation condition between ID and OOD errors presented in Section 3.2.

**Proposition 1.** *Let bias parameter $b$ follow a half-normal distribution with scale parameter $k$, and let error components be defined as $T(b) = b^2$ and $U(b) = (b - \Delta)^2$. The Pearson correlation coefficient between $T(b)$ and $U(b)$ is negative if and only if $k < \alpha\Delta$.*

*Proof.* The moment generating function of the half-normal distribution with scale parameter $k$ is:

$$M(t) = \exp\left(\frac{k^2 t^2}{2}\right)\left[1 + \text{erf}\left(\frac{kt}{\sqrt{2}}\right)\right] \tag{13}$$

From this, we derive the raw moments:

$$\mathbb{E}[b^n] = \begin{cases} \alpha k^n (n-1)!!, & \text{if } n \text{ is odd} \\ k^n (n-1)!!, & \text{if } n \text{ is even} \end{cases} \tag{14}$$

For our analysis, we need the first four moments:

$$\mathbb{E}[b] = \alpha k, \quad \mathbb{E}[b^2] = k^2, \quad \mathbb{E}[b^3] = 2\alpha k^3, \quad \mathbb{E}[b^4] = 3k^4 \tag{15}$$

The Pearson correlation coefficient is defined using expectations as:

$$\rho_{X,Y} = \frac{\text{Cov}(X,Y)}{\sigma_X \sigma_Y} = \frac{\mathbb{E}[(X - \mu_X)(Y - \mu_Y)]}{\sqrt{\mathbb{E}[(X - \mu_X)^2]\mathbb{E}[(Y - \mu_Y)^2]}} \tag{16}$$

where $\mu_X = \mathbb{E}[X], \mu_Y = \mathbb{E}[Y]$ are the means, $\sigma_X = \sqrt{\mathbb{E}[(X - \mu_X)^2]}, \sigma_Y = \sqrt{\mathbb{E}[(Y - \mu_Y)^2]}$ are the standard deviations, and $\text{Cov}(X,Y) = \mathbb{E}[(X - \mu_X)(Y - \mu_Y)]$ is the covariance. For our specific case with $T(b) = b^2$ and $U(b) = (b - \Delta)^2$, after simplifying, the covariance between $T(b)$ and $U(b)$ is:

$$\text{Cov}(T,U) = 2k^3(k - \alpha\Delta) \tag{17}$$

The Pearson correlation coefficient is:

$$\rho_{T,U} = \frac{k - \alpha\Delta}{\sqrt{k^2 - 2\alpha\Delta k + \left(2 - \frac{4}{\pi}\right)\Delta^2}} \tag{18}$$

Since the denominator is always positive for $k > 0$, the correlation $\rho_{T,U}$ is negative precisely when $k < \alpha\Delta$. This negative correlation condition defines the regime where deliberately increasing ID bias improves OOD performance. The correlation $\rho_{T,U}$ is strictly increasing in $k$ throughout $(0, +\infty)$. As $k$ increases from its minimal feasible positive values toward $\infty$, $\rho_{T,U}$ increases from approximately $-0.936$ to $1$, with the strongest negative correlation achieved at the smallest feasible values of $k > 0$. $\square$

## B   EXPERIMENTAL PARAMETER DETAILS

This appendix provides detailed parameter information for our experimental setup, including both the distribution shift level thresholds and the $W_1$ distances between distributions.

### B.1   DISTRIBUTION SHIFT LEVEL PARAMETERS

Permutations in our experiments were categorized into three levels based on their deviation patterns from the global training distribution. Table 3 shows the threshold values used for this categorization.

Permutations are categorized as Low if outlier count $\leq \tau_{\text{low}}$, Medium if $\tau_{\text{low}} < \text{count} \leq \tau_{\text{high}}$, and High if count $> \tau_{\text{high}}$.

Table 3: Distribution shift level thresholds for both computational approaches

| Approach | Dataset | Threshold | $\tau$ Value | Probability |
|---|---|---|---|---|
| Batchwise | T1S1 | $\tau_{low}$ | 3 | 0.35 |
| | | $\tau_{high}$ | 6 | 0.31 |
| | QM9 | $\tau_{low}$ | 3 | 0.29 |
| | | $\tau_{high}$ | 6 | 0.32 |
| | MSD | $\tau_{low}$ | 2 | 0.14 |
| | | $\tau_{high}$ | 7 | 0.16 |
| Cumulative | T1S1 | $\tau_{low}$ | 3 | 0.68 |
| | | $\tau_{high}$ | 13 | 0.11 |
| | QM9 | $\tau_{low}$ | 3 | 0.71 |
| | | $\tau_{high}$ | 13 | 0.13 |
| | MSD | $\tau_{low}$ | 6 | 0.76 |
| | | $\tau_{high}$ | 14 | 0.11 |

## B.2 DATASET DISTRIBUTION SHIFT CHARACTERISTICS

To quantify the magnitude of distribution shifts in our experiments, we measured the $W_1$ distances in label space between training and test distributions, as shown in Table 4.

Table 4: $W_1$ distances in label space for ID and OOD settings

| Dataset | ID Distance | OOD Distance |
|---|---|---|
| T1S1 | 0.02 | 0.77 |
| QM9 | 0.01 | 0.61 |
| MSD | 0.01 | 0.86 |

The significant difference between ID and OOD distances confirms the presence of meaningful distribution shifts in our experimental setup. We maintained OOD distances greater than 0.6 to ensure distribution shifts were substantial enough for evaluation, while keeping ID distances lower to reflect standard cross-validation scenarios. All measurements were computed after separate normalization of training and test sets, which reflects realistic deployment scenarios where test distribution parameters are not available during training.

## B.3 DETAILED PERFORMANCE BY DEVIATION GROUP

Table 5 provides a detailed breakdown of the performance (MAE and RMSE) for each deviation group identified by the ADB framework on the T1S1 dataset. This data supports the analysis presented in Section 5.2, illustrating the performance differences that arise from varying levels of induced distributional diversity during training. Similar patterns can be observed for the QM9 dataset in Table 6 and the MSD dataset in Table 7.

## B.4 STATISTICAL SIGNIFICANCE ANALYSIS

To validate the statistical significance of our performance improvements, we conducted paired t-tests comparing the ADB approach against traditional cross-validation. Table 8 presents the p-values resulting from these tests across all datasets and computational approaches. Results with $p < 0.05$ indicate statistically significant performance differences, providing strong evidence that the observed improvements are not due to random variation.

We assessed statistical significance using 10-fold cross-validation with fixed random seeds, analyzing OOD errors across all folds. This methodology accounts for fold-specific variations while ensur-

Table 5: Performance comparison across deviation groups on T1S1

| Approach | Deviation | Metric | Value | PR (%) |
|---|---|---|---|---|
| Batchwise | Low | MAE | $0.436 \pm 0.011$ | 73.3 |
| | | RMSE | $0.535 \pm 0.010$ | 68.9 |
| | Medium | MAE | $0.466 \pm 0.010$ | 48.9 |
| | | RMSE | $0.564 \pm 0.009$ | 38.9 |
| | High | MAE | $0.426 \pm 0.008$ | 81.1 |
| | | RMSE | $0.519 \pm 0.007$ | 81.1 |
| Cumulative | Low | MAE | $0.403 \pm 0.004$ | 87.6 |
| | | RMSE | $0.496 \pm 0.005$ | 87.1 |
| | Medium | MAE | $0.397 \pm 0.009$ | 88.8 |
| | | RMSE | $0.490 \pm 0.007$ | 90.6 |
| | High | MAE | $0.436 \pm 0.004$ | 57.3 |
| | | RMSE | $0.530 \pm 0.003$ | 60.0 |

Table 6: Performance comparison across deviation groups on QM9

| Approach | Deviation | Metric | Value | PR (%) |
|---|---|---|---|---|
| Batchwise | Low | MAE | $0.149 \pm 0.001$ | 57.8 |
| | | RMSE | $0.194 \pm 0.002$ | 51.1 |
| | Medium | MAE | $0.137 \pm 0.002$ | 84.4 |
| | | RMSE | $0.194 \pm 0.001$ | 50.0 |
| | High | MAE | $0.135 \pm 0.002$ | 86.7 |
| | | RMSE | $0.193 \pm 0.003$ | 55.6 |
| Cumulative | Low | MAE | $0.150 \pm 0.002$ | 48.9 |
| | | RMSE | $0.200 \pm 0.002$ | 26.7 |
| | Medium | MAE | $0.134 \pm 0.002$ | 82.2 |
| | | RMSE | $0.179 \pm 0.002$ | 74.4 |
| | High | MAE | $0.136 \pm 0.002$ | 73.3 |
| | | RMSE | $0.181 \pm 0.002$ | 72.2 |

ing fair comparison between methods, confirming that our performance improvements are consistent across multiple datasets and evaluation metrics.

Table 7: Performance comparison across deviation groups on MSD

| Approach | Deviation | Metric | Value | PR (%) |
|---|---|---|---|---|
| Batchwise | Low | MAE | $0.860 \pm 0.001$ | 60.5 |
| | | RMSE | $1.153 \pm 0.001$ | 53.6 |
| | Medium | MAE | $0.856 \pm 0.001$ | 79.0 |
| | | RMSE | $1.148 \pm 0.002$ | 71.4 |
| | High | MAE | $0.860 \pm 0.001$ | 60.5 |
| | | RMSE | $1.158 \pm 0.001$ | 32.1 |
| Cumulative | Low | MAE | $0.848 \pm 0.001$ | 95.0 |
| | | RMSE | $1.144 \pm 0.007$ | 83.5 |
| | Medium | MAE | $0.859 \pm 0.001$ | 71.3 |
| | | RMSE | $1.148 \pm 0.001$ | 77.2 |
| | High | MAE | $0.863 \pm 0.001$ | 62.5 |
| | | RMSE | $1.152 \pm 0.001$ | 63.3 |

Table 8: Statistical significance (p-values from paired t-tests) of ADB compared to CV

| Method | Dataset | Metric | p-value | Significant ($p < 0.05$) |
|---|---|---|---|---|
| Batchwise | T1S1 | MAE | $1.4 \times 10^{-5}$ | Yes |
| | | RMSE | $5.2 \times 10^{-5}$ | Yes |
| | QM9 | MAE | $2.8 \times 10^{-4}$ | Yes |
| | | RMSE | $2.5 \times 10^{-2}$ | Yes |
| | MSD | MAE | $5.6 \times 10^{-5}$ | Yes |
| | | RMSE | $3.7 \times 10^{-4}$ | Yes |
| Cumulative | T1S1 | MAE | $< 1.0 \times 10^{-6}$ | Yes |
| | | RMSE | $6.0 \times 10^{-6}$ | Yes |
| | QM9 | MAE | $< 1.0 \times 10^{-6}$ | Yes |
| | | RMSE | $2.0 \times 10^{-4}$ | Yes |
| | MSD | MAE | $< 1.0 \times 10^{-6}$ | Yes |
| | | RMSE | $< 1.0 \times 10^{-6}$ | Yes |