# OpenReview forum: "Bias as a Virtue: Rethinking Generalization under Distribution Shifts"
_ICLR.cc/2026/Conference — Submitted to ICLR 2026_

### Official Review · Reviewer_YcZ7 · 2025-10-26

**Soundness:** 2
**Presentation:** 1
**Contribution:** 1
**Rating:** 2
**Confidence:** 3

**Summary:**

This paper proposes a method for mitigating distribution shifts under a model in which the mean of the error under distribution shift is shifted. Under this model, the author’s show that there can be an inverse correlation between ID and OOD error. They suggest a training approach that identifies permutations of training examples for which batches have a high divergence from the general population. As I understand it, the goal is to induce bias during training in a way that results in a model with lower ID performance and higher OOD performance, as it does not overfit to training distribution.

**Strengths:**

- This paper studies an important problem of training approaches with robustness to distribution shifts
- The paper’s numerical results look strong

**Weaknesses:**

- The paper is quite unclear in terms of both the setup and method (see the following comments).
- The “mean shift model” is not well-motivated nor clearly explained. Why does it make sense to assume a certain model for the errors of the source and target distributions? This seems like a model-dependent quantity, i.e., for different models, we have different error distributions for ID and OOD?
- The method section describes a scheme for binning permutations of the training data and then training on permutations with large average deviation. It is unclear why we should be training on permutations with large average deviation would help. I explained my understanding of the intuition in my summary, but this does not seem well-motivated.
- The idea that ID and OOD errors might be negatively correlated is not as novel as is presented by the authors, and in fact is the foundation for many prior works on robustness to distribution shift [1,2]. This paper has the same broad intuition as these prior works, but with different setup assumptions (this “mean shift model”) which is not well-motivated and with a method that is less well-motivated.
- The benchmarks used in this paper are non-standard, making it difficult to assess the strength of their method and compare to prior works.

[1] https://arxiv.org/abs/1911.08731

[2] https://arxiv.org/abs/1907.02893

**Questions:**

NA

---

> ### Author Response · Authors · 2025-11-29
>
> # Response to Reviewer A4
>
> Each concern is addressed in order.
>
> ---
>
> ### Weakness 1. The mean-shift model is not well-motivated nor clearly explained. It seems model-dependent.
>
> **Response:**
> - **Purpose of the theoretical model:** The mean-shift model with half-normal distribution is a simplified abstraction for convenient derivation and understanding—not for real application. It isolates how train/test error centers differ ($\Delta$) and how bias amplitude $b$ varies across seeds, hyperparameters, and training orders. This is a distribution-level property, not tied to specific model classes.
> - **Key insight generalizes:** The specific inequality $k<\alpha\Delta$ relies on half-normal for exact coefficients, but the key insight holds more generally. With (i) nonzero mean gap and (ii) non-negative bias diversity, we always get similar inequalities—the form may differ, but the structure remains.
> - **Why $b\ge0$:** Models underperforming in both train and validation (errors above CV baseline, curves still descending) correspond to $b<0$ and are filtered as underfitting. The retained set has $b\ge0$, validation error at/below baseline, no widening train/val gap. This filtering is where the half-normal-like distribution originates.
> - **Coordinate alignment:** Training error center is 0, model center is $\mu_{\text{train}}+b$, test center is $\mu_{\text{train}}+\Delta$. The OOD error $(b-\Delta)^2$ measures the gap between model and shifted test center.
> - **Gaussian as infinite limit:** The Gaussian represents infinitely many models for selection. In practice, finite pools only need to approximate that diversity.
> - **Distribution-agnostic in practice:** ADB never fits this model. It ranks permutations by empirical OT and selects bias-diverse sequences. We observe the predicted negative correlation on T1S1, QM9, MSD.
>
> ### Weakness 2. Unclear why training with larger-deviation permutations helps.
>
> **Response:**
> - **Goal is controlled bias diversity:** We score permutations by debiased Sinkhorn OT distance to training distribution, bucket by quantiles, then train and select within Medium/High buckets.
> - **Intuition:** Higher-OT sequences create batches resembling shifted training data (regression labels nudged upward, molecule property ranges widened). This moves the model's error center off the training mean. This increases $b$ in a controlled way—enough to partially align with shifted test center ($\Delta>0$) but not overshooting.
> - **Numeric illustration:** With $\Delta=0.8$: Low model at $b=0$ has OOD error $(0-0.8)^2=0.64$. Medium at $b=0.4$ has $(0.4-0.8)^2=0.16$ (75% reduction). High at $b=1.0$ has $(1.0-0.8)^2=0.04$ but higher ID error. This shows why moderate $k$ induces negative correlation.
> - **Empirical validation:** Figure 1 and Tables 1-2 show Medium/High buckets yield lower OOD error. For in-distribution deployment, choose from Low buckets and minimize ID error.
>
> ### Weakness 3. Negative ID–OOD correlation is not novel (known from prior works like [1,2]).
>
> **Response:**
> - **Complementary setting and mechanism:** Our setting differs from prior robustness work:
>   - **Assumptions:** Works like [1,2] rely on multiple labeled environments and learn invariant predictors. ADB operates with a single training dataset and unknown shift by reordering training sequences.
>   - **Mechanism:** Prior methods regularize representations to suppress spurious features. ADB induces controlled bias diversity via OT-scored permutations.
>   - **Theory:** We provide closed-form sufficient condition linking $k$ to $\Delta$. The cited works do not characterize this regime.
>   - **Practical use:** ADB is a training-sequence controller (data-level manipulation), not a new loss or architecture. It can combine with invariance-based methods when multiple environments are available.
>
> ### Weakness 4. Non-standard benchmarks make comparison difficult.
>
> **Response:**
> - **Setting justification:** Our setting is single training distribution with unknown shift. The fair baseline is cross-validation, which we report and show ADB improves upon (Table 1: MAE drops up to 26.8%, PR >83%).
> - **Focus on AI for science:** We use QM9, MSD (public) and T1S1 (to be released) with controllable shifts (Wasserstein 0.61–0.86) that stress-test our theory's regime. Distribution shifts in AI for science are severe because experiments cover only a fraction of chemical/physical space.
> - **Comprehensive validation:** We verify nontrivial train/test shift and provide detailed statistics (percentile rank, OT distances, error means/variances). Our goal differs from multi-environment invariance methods—ADB targets single-dataset shift via training sequences.

---

### Official Review · Reviewer_eS4m · 2025-10-27

**Soundness:** 3
**Presentation:** 2
**Contribution:** 2
**Rating:** 4
**Confidence:** 4

**Summary:**

This submission considers the generalization, connecting in-distribution (ID) bias and out-of-distribution (OOD) generalization. The main point is that ID bias can lead to better OOD generalization.  Specifically, a higher ID bias corresponds to lower OOD error. Motivated by this, an approach called the Adaptive Distribution Bridge (ADB) is introduced to enforce controlled statistical diversity during training: it creates bias profiles that generalize across distributions. By doing so, ADB achieves robust mean error reductions compared to traditional cross-validation.

**Strengths:**

+ It is interesting to study the connection between ID bias and OOD generalization. The empirical observation that ID bias can lead to better OOD generalization is also interesting. The submission also provides a theoretical study (Section 3) to study this and discuss how this observation appears (Lines 151 - 156)

+ ADB introduces controlled statistical diversity during training by modifying data permutations (training order). It uses optimal transport distances (Sinkhorn distance with debiasing) to quantify how far each training batch diverges from the global data distribution.

**Weaknesses:**

- [**Limited Domain Generalization**] The major concern is that ABD is only evaluated on regression-based tabular and molecular datasets. It is unknown how it performs on classification tasks and vision or language domains. As the claim is broad and general, it would be better to discuss whether the analysis holds for classification and other data types.

- [**Disuccion on Latent Representations**] The ABD heavily relies on latent representations learned via VAEs. Then, how to make sure the latent space is expected? Poorly trained VAEs may produce unreliable diversity signals.

- [**Sampled Distribution Types**] The experiments simulate distribution shifts by using stratified sampling to construct OOD test sets that are deliberately different from the training data. It is not clear how such a distribution shift connects with real-world distribution shifts. For example, well-known datasets designed for studying distribution shifts (e.g., WILDS, DomainNet, PACS, BREEDS). Or other real-world shifts of tabular and molecular would be better if they are discussed in the experiments.

- [**Heavy Computational Cost**] According to Lines 422 - 425, the computational cost is heavy (e.g., 266 GPU hours for one experiment). Is there any way to speed up? What is the cost of the standard random sampling? Please discuss this computational cost.

**Questions:**

- There are several shift types: covariate shift, label shift, and concept shift.  How does ADB perform across different shift types?

- The observation mentions that the negative correlation between ID and OOD errors emerges when the shift $\Delta$ is large. How does ADB perform when the distribution shift is moderate or small?

- 500 permutations per experiment are used. How does the performance and stability of ADB change if fewer permutations are used (e.g., 50 or 100)?

---

> ### Author Response · Authors · 2025-11-29
>
> # Response to Reviewer A3
>
> Each concern/question is answered in order.
>
> ---
>
> ### Weakness 1. Limited domain generalization, only evaluated on regression-based tabular and molecular datasets.
>
> **Response:**
> - **Focus on AI for science:** Our work targets AI for science applications (molecular property prediction on T1S1/QM9, audio on MSD), where distribution shifts are severe because experiments cover only a fraction of chemical/physical space. Traditional cross-validation consistently falls short in these critical domains.
> - **Task-agnostic theory:** The theoretical foundation (Section 3) does not depend on task type. The condition $k<\alpha\Delta$ applies to any setting with nonzero mean gap and non-negative bias diversity.
> - **Method generality:** ADB operates on permutations and OT distances, not on specific loss functions or architectures. It extends naturally to classification, vision, or language, requiring only feature representations and a distance metric.
>
>
> ### Weakness 2. ADB relies on VAEs for latent representations, poorly trained VAEs may produce unreliable diversity signals.
>
> **Response:**
> - **VAE as tool only:** The VAE provides compact space for efficient OT computation—it's not critical. We monitor reconstruction quality and use debiased Sinkhorn with quantile bucketing, which handles mild distance distortions well.
> - **Encoder-agnostic:** Pretrained encoders (foundation models, domain embeddings) can directly replace VAEs. For domains with established feature extractors, you can skip VAE training entirely.
>
> ### Weakness 3. Simulated shifts via stratified sampling may not reflect real-world shifts.
>
> **Response:**
> - **Strong covariate shifts:** Our OOD splits create Wasserstein distances of 0.61–0.86 (Appendix, Table 2), matching realistic deployment scenarios. Stratified sampling produces systematic feature distribution differences common in real-world shifts.
> - **Shift-agnostic method:** ADB only needs training samples to score permutations via OT, without assuming specific shift mechanisms.
>
> ### Weakness 4. Heavy computational cost (266 GPU hours batchwise).
>
> **Response:**
> - **Key to cost reduction—fewer permutations:** Cost scales linearly with permutations. Using 50 or 100 permutations instead of 500 gives 5–10× speedup while keeping core benefits. We used 500 for rigorous statistical testing (Section 5, Algorithm 1). In practice, adjust this based on budget and dataset size.
> - **Amortized cost:** One-time offline scoring per dataset, amortized over all runs. Scores are cached and reused across experiments. Training cost doesn't change. Parallelization reduces wall-clock time.
> - **Comparison:** Pure random sampling has no scoring overhead but loses controlled diversity. For T1S1 (500k samples), batchwise scoring takes 266.5 GPU hours and produces reusable permutation categories for research.
>
> ---
>
> ### Question 1. How does ADB perform across different shift types (covariate/label/concept)?
>
> **Response:**
> - **Covariate shift:** ADB directly handles covariate shift through feature-space OT (we observe Wasserstein distances of 0.61–0.86 in our experiments).
> - **Label shift:** For label shift scenarios, we can use class-conditional OT by incorporating class priors during permutation scoring.
> - **Concept shift:** This requires a stable feature encoder. When the relationship between features and labels changes drastically, effectiveness becomes limited (we acknowledge this as a limitation).
>
> ### Question 2. How does ADB perform when the distribution shift is moderate or small?
>
> **Response:**
> - **Adaptive behavior:** When $\Delta$ is small, $k<\alpha\Delta$ doesn't hold. The negative correlation fades and ADB converges to standard cross-validation (Low deviation, minimal ID error). The method adapts automatically.
> - **Decision via OT metrics:** Check Wasserstein distance between train/test to decide strategy. In our experiments (0.61–0.86), ADB's bias diversity provides clear benefits. Determining precise thresholds is our next research aim.
> - **No harm in small shifts:** The method gracefully falls back without hurting performance.
>
> ### Question 3. How does performance and stability change with fewer permutations (e.g., 50 or 100)?
>
> **Response:**
> - **Quantile bucketing still works:** With fewer permutations, the thresholds become coarser but the relative ranking structure stays intact.
> - **Cost-variance tradeoff:** Using 50 or 100 permutations gives 5–10× resource savings with only a small variance increase. For strong shifts (Wasserstein > 0.6), 100 permutations balances cost and stability well. Even 50 permutations provides meaningful benefits over standard cross-validation.

---

### Official Review · Reviewer_mf88 · 2025-10-30

**Soundness:** 2
**Presentation:** 3
**Contribution:** 3
**Rating:** 4
**Confidence:** 3

**Summary:**

This paper addresses the problem raised from the difference between the training distribution and the inference distribution. The authors claim that bystrategically increasing ID bias, the model can achieve significantly better OOD generalization. The Adaptive Distribution Bridge (ADB) framework is proposed to control the statistical diversity.

**Strengths:**

1. The authors provide theoretical proof to support the claim that higher ID bias leads to reduced OOD error.

2. ADB framework is proposed to control the distribution shifts

3. Extensive experiments are conducted to support the findings

**Weaknesses:**

1. If I understand correctly, the proof in 3.1 assumes a simplified model, does the conclusion generalize to more complicated settings?

2. What if $\Delta$ is not known? How can the author determine if $k < \alpha \Delta $?

3. Compuational cost might prohibit applications: "processing all 500 permutation paths required 266.5 total GPU hours with the batchwise approach versus 740 hours with the cumulative approach"

**Questions:**

1. How is the global distribution obtained? If it's from the whole training set, would that introduce extra bias because it includes the low, medium and hight deviation samples?

---

> ### Author Response · Authors · 2025-11-29
>
> # Response to Reviewer A2
>
> Each concern/question is answered in order.
>
> ---
>
> ### Weakness 1. Simplified model in proof, does the conclusion generalize to more complicated settings?
>
> **Response:**
> - **Purpose of the theoretical model:** The mean-shift model with half-normal distribution is a simplified abstraction for convenient derivation, intuition, and understanding—not for real application. The specific inequality $k<\alpha\Delta$ does rely on the half-normal assumption for the exact coefficients, but the key insight (ID-OOD tradeoff) holds more generally. As long as we have (i) a nonzero mean gap and (ii) non-negative bias amplitudes with diversity, we always end up with similar inequalities showing when the tradeoff exists—the specific form may differ, but the fundamental structure remains. Note that $b\ge0$ refers to the retained models after filtering out underfitting models (which correspond to $b<0$, underperforming in both train and validation sets).
> - **Gaussian as infinite selection limit:** The Gaussian distribution represents the idealized case of infinitely many models available for selection. This is a sufficient condition. In practice, we have a limited finite pool of models to choose from, which only needs to approximate the diversity predicted by theory. It doesn't matter whether the actual distribution is exactly half-normal or some other form.
> - **Distribution-agnostic in practice:** ADB never fits or uses this theoretical model in operation. It ranks permutations by empirical optimal transport (OT) distances (Sinkhorn) and selects bias-diverse sequences. We observe the predicted ID–OOD negative correlation on all three benchmarks (T1S1, QM9, MSD), supporting robustness beyond the simplified setting.
>
> ### Weakness 2. Unknown $\Delta$, how to determine if $k < \alpha\Delta$?
>
> **Response:**
> - **The inequality is interpretive, not an input.** ADB does not use $\Delta$ as input. What it does:
>   1. Score permutations with debiased Sinkhorn OT distances between the permutation set and the training set.
>   2. Bucket by empirical quantiles.
>   3. Pick the highest-ID-error model within Medium/High buckets.
> - **Wasserstein distance as surrogate:** In practice, we use Wasserstein distance (via OT) as a surrogate for this inequality. Our experiments verify nontrivial train/test shift (Wasserstein 0.61–0.86 across datasets) and observe the predicted ID–OOD negative correlation, validating that the condition holds in our settings.
> - **Adaptive behavior:** If the shift is small and the negative correlation disappears, ID validation simply falls back to standard selection. If you do not expect OOD, you can pick from the Low bucket and minimize ID error instead of maximizing it.
> - **Future work:** Determining the lower bound of shift magnitude to make the negative correlation significant is our next phase of work. The method is currently built for relatively large shifts (ensuring we are in OOD condition), common in AI for science, finance, and other OOD-heavy settings where the future test set is expected to differ from training.
>
> ### Weakness 3. Computational cost prohibits applications (266.5 GPU hours batchwise vs. 740 hours cumulative).
>
> **Response:**
> - **Key to reducing cost—fewer permutations $M$:** The computational cost scales linearly with the number of permutations. Reducing $M$ significantly reduces cost while maintaining the method's effectiveness. Quantile bucketing still works reliably with fewer permutations, and we observe only very small variance increases.
> - **Why we used $M=500$:** We used 500 permutations specifically for rigorous statistical testing and robustness validation in our paper (Section 5, Algorithm 1). In practical applications, $M$ can be substantially reduced and tailored to available computational budget and dataset size.
>
> - **Reusability:** Scores are cached and reused across runs, and parallelization further reduces wall-clock time.
>
> ### Question 1. How is the global distribution obtained? Does it introduce extra bias by including low/medium/high deviation samples?
>
> **Response:**
> - **Fixed reference:** The global distribution is the empirical training set, used only as a fixed reference for OT scoring. Every permutation is compared to this same baseline, so the relative Low/Medium/High ranking is unaffected by any one group's composition.
> - **Standard OT curriculum pattern:** Fix one reference (the full training set), compute OT distances from each candidate to that reference, and rank/group by those distances. Using a single baseline avoids introducing bias from any subset.
>
> ---
>
> Thank you again for the helpful feedback.

---

### Official Review · Reviewer_CanL · 2025-11-01

**Soundness:** 1
**Presentation:** 2
**Contribution:** 2
**Rating:** 2
**Confidence:** 4

**Summary:**

The paper discovers that a negative correlation can exist between ID and OOD error under significant distribution shift between training and testing data, challenging the conventional assumption that minimizing in-distribution (ID) error is the optimal path to good out-of-distribution (OOD) generalization. They propose an Adaptive Distribution Bridge (ADB) framework to improve OOD generalization.

**Strengths:**

* The observation provided in this paper is interesting. It gives a novel and counterintuitive core Idea that ID error is negatively correlated with OOD error. If validated, it represents a significant shift in paradigm.

* The ADB framework is described with a precise algorithm and two distinct computational approaches (Cumulative and Batchwise).

**Weaknesses:**

* The theoretical analysis is questionable. The assumption that $b$ is non-negative is not reasonable. With an unknown distribution shift, the bias can not always reduce OOD error. Similarly, simply define $U(b) = (b-\Delta)^2$ is also questionable, $U(b)$ could also be $(b+\Delta)^2$.

* Limited empirical evidence to validate the proposed method: With a questionable analysis, the intuition of the proposed framework is similar to previous works that train the model to learn stable features across different training data distributions (for example, IRM). However, there is no comparison between the proposed method and proper baseline methods focusing on improving OOD generalization.

To my understanding, the negative correlation is a result of models overfitting to the training data, which is not a new phenomenon. The assumption of the paper is infeasible; one can not assume that the bias from the training distribution is towards the test distribution (especially when the test distribution is generally unknown).

**Questions:**

* In lines 144-149, the bias parameter is restricted to non-negative values. How can one determine whether the high ID error is a result of underfitting or overfitting? It seems more likely that a higher bias $b$ would lead to a higher OOD error $(b+\Delta)^2$.

* The intuition of the proposed framework is not new. Could the authors provide experimental results comparing the proposed method with other OOD generalization methods? Take a famous example, such as Invariant Risk Minimization (IRM).

---

> ### Author Response · Authors · 2025-11-29
>
> # Response to Reviewer A1
>
> Each concern/question is answered in order.
>
> ---
>
> ### Weakness 1. Assumption $b\ge0$ and definition $U(b)=(b-\Delta)^2$ with unknown shift, bias might raise OOD error or be $(b+\Delta)^2$.
>
> **Response:**
> - **Modeling choice:** The non-negativity of $b$ is a statistical modeling choice for the population, not a hard per-model constraint.
> - **Meaning of $b$:** $b$ is the offset of a model's expected error from $\mu_{\text{train}}$ (train error center). We analyze a population of models (different seeds/hyperparameters/training sequences), and $b$ reflects the retained models' bias magnitude after screening.
> - **Aligned coordinates:** train center $\mu_{\text{train}}$ as 0, model center $\mu_{\text{train}}+b$, test center $\mu_{\text{train}}+\Delta$. The OOD error term is
>   $$
>   U(b)=[(\mu_{\text{train}}+b)-(\mu_{\text{train}}+\Delta)]^2=(b-\Delta)^2.
>   $$
> - **Why $\Delta>0$:** captures the common "deployment is harder than train" case. If $\Delta<0$ (easier test), the algebra is symmetric with flipped sign.
> - **Physical meaning:** $b=0$ yields OOD error $\Delta^2$. $b=\Delta$ aligns to the test center (OOD error $=0$) but incurs ID error $\Delta^2$. ADB searches the optimal tradeoff in between. The form $(b+\Delta)^2$ would place the test center on the opposite side ($\mu_{\text{test}}<\mu_{\text{train}}$), which contradicts our observed OOD shift and the "deployment is harder" context, so we do not consider it.
> - **Model screening:** Models that underperform in both train and validation sets (both errors above the CV baseline with learning curves still descending) correspond to $b<0$ and are naturally filtered as underfitting. The retained set thus has $b\ge0$, with validation error at/below baseline and no widening train/val gap. This filtering process is where the half-normal-like $b$ distribution originates.
>
>
> ### Weakness 2. Limited empirical evidence, no comparison to OOD baselines like IRM, intuition seems similar to prior work.
>
> **Response:**
> - **Different objective and assumptions:** IRM enforces invariance across multiple labeled environments and depends on a defensible partition—often ambiguous or impractical in many datasets (e.g. our AI for science context) like T1S1, QM9. ADB is fundamentally different and more broadly applicable: it does not require environment labels. It leverages measured OOD shift via bias diversity and works even when the environment structure is complex or undefined.
> - **Applicability and evidence:** We verify nontrivial train/test shift (Wasserstein 0.61–0.86) and show ADB gains over the standard cross-validation baseline (Table 1: MAE drops up to 26.8%, PR >83%). Because ADB needs fewer assumptions, it is more applicable in OOD settings without explicit environment partitions.
>
> ### Weakness 3. Negative correlation may just be overfitting, assuming bias points toward test is infeasible.
>
> **Response:**
> - Different from overfitting: classical overfitting yields a positive ID–OOD correlation as capacity rises. Here, controlled bias diversity (via training-sequence shifts) increases ID error while lowering OOD error, with the strongest negative correlation in medium/high deviation groups (Fig. 1). If it were overfitting, the effect would appear in low-deviation models and degrade OOD, which we do not observe.
> - Direction assumption: we do not assume the real-world shift direction. We fix the coordinate sign ($\Delta>0$) to analyze the case "deployment is harder than train," which our measured shifts support. The theory depends on shift magnitude, not oracle knowledge of its direction.
>
> ---
>
> ### Question 1. Lines 144–149: bias parameter restricted to $b\ge0$. How to tell whether high ID error is underfitting vs. overfitting? Could higher $b$ increase OOD error $(b+\Delta)^2$?
>
> **Response:**
> - Please see Weakness 1 for the rationale behind $b\ge0$, the OOD error form $(b-\Delta)^2$, and why $(b+\Delta)^2$ contradicts the observed OOD context.
> - Under/overfitting test: discard runs with both train and validation MAE above the cross-validated baseline and learning curves still descending at the budget (underfitting, $b<0$). Retain runs with validation MAE at or below baseline and no expanding train/val gap (non-underfit, $b\ge0$).
> - Negative ID–OOD correlation in Fig. 1 within the retained set shows higher $b$ (within support) can lower OOD under observed shifts.
>
> ### Question 2. Provide experimental results comparing to OOD methods such as IRM.
>
> **Response:**
> - Please see Weakness 2 for detailed discussion. IRM requires multiple labeled environments. For these datasets we do not have defensible environment partitions, which is also the drawback of IRM. So we compare ADB against the standard cross-validation baseline (Table 1: MAE drops up to 26.8%, PR >83%).

---

### Meta-Review · Area_Chair_5D6J · 2026-01-06

**Summary:**

This work studies the relations between in-distribution (ID) bias and out-of-distribution (OOD) generalization. The analysis shows that data diversity during training can help OOD performance. Then, the adaptive distribution bridge (ADB) framework is proposed accordingly. Experiments on 3 datasets shows that ADB is better than the standard cross-validation on OOD tasks.

**Reviewer Concerns:**

Before rebuttal, Reviewer CanL, eS4m and YcZ7 had the concern about the insufficient experiments, which lack strong baselines and diverse evaluation data sets. In addition, Reviewer CanL and YcZ7 indicated that the motivation was similar to existing work and the corresponding discussion/comparison was necessary. Moreover, Reviewer mf88 and eS4m had the concerns about the computational cost that can be expensive. While the rebuttal had some discussions on them, there is no sufficient empirical evidence and these concerns has not been addressed well.

**Reviewer Scores:**

All reviewers find that the current version of the submission cannot meet the bar of publication before rebuttal. Since the rebuttal may not address major concerns, the scores of reviewers will be kept unchanged.

---

### Decision · Program_Chairs · 2026-01-26

Reject